# COVID-19 Pandemic Impacted Food Security and Caused Psychosocial Stress in Selected States of Nigeria

**DOI:** 10.3390/ijerph20054016

**Published:** 2023-02-23

**Authors:** Dauda G. Bwala, Olutosin A. Otekunrin, Oluwawemimo O. Adebowale, Modupe M. Fasina, Ismail A. Odetokun, Folorunso O. Fasina

**Affiliations:** 1Virology Department, National Veterinary Research Institute, Vom 930101, Nigeria; 2Department of Agricultural Economics and Farm Management, Federal University of Agriculture, Abeokuta 110124, Nigeria; 3Department of Veterinary Public Health and Preventive Medicine, College of Veterinary Medicine, Federal University of Agriculture, Abeokuta 110124, Nigeria; 4Institute of Tropical Medicine and International Health, Charité—Universitätsmedizin Berlin, 10117 Berlin, Germany; 5Department of Veterinary Public Health and Preventive Medicine, Faculty of Veterinary Medicine, University of Ilorin, Ilorin 240272, Nigeria; 6Food and Agriculture Organization of the United Nations, Nairobi 00601, Kenya; 7Department of Veterinary Tropical Diseases, University of Pretoria, Pretoria 0110, South Africa

**Keywords:** COVID-19, food insecurity, psychological impact, socio-economics, food access, Nigeria

## Abstract

The COVID-19 disease has infected many countries, causing generalized impacts on different income categories. We carried out a survey among households (n = 412) representing different income groups in Nigeria. We used validated food insecurity experience and socio-psychologic tools. Data obtained were analyzed using descriptive and inferential statistics. The earning capacities of the respondents ranged from 145 USD/month for low-income earners to 1945 USD/month for high-income earners. A total of 173 households (42%) ran out of food during the COVID-19 pandemic. All categories of households experienced increasing dependency on the general public and a perception of increasing insecurity, with the high-income earners experiencing the greatest shift. In addition, increasing levels of anger and irritation were experienced among all categories. Of the socio-demographic variables, only gender, educational level of the household head, work hours per day, and family income based on society class were associated (*p* < 0.05) with food security and hunger due to the COVID-19 pandemic. Although psychological stress was observed to be greater in the low-income earning group, household heads with medium and high family income were more likely to have satisfactory experiences regarding food security and hunger. It is recommended that socio-economic groups should be mapped and support systems should target each group to provide the needed support in terms of health, social, economic, and mental wellness.

## 1. Introduction

In December 2019, an influenza-like illness, later designated as COVID-19 and caused by SARS-CoV-2 virus, was first reported in Wuhan, China [1,2,3]. Since the time of this first report, COVID-19 has spread to infect at least 213 countries and territories globally, and nearly 755.39 million cases have been reported in humans, with associated human deaths in excess of 6.83 million by 12 February 2023 [4]. Based on observations, the disease is not only a public health issue but remains multi-dimensional, with numerous other issues including but not limited to disruptions to livelihoods and employment, psychosocial consequences, endemic hungers, and poverty [5,6,7,8]. While detailed field reports have documented a number of consequences associated with COVID-19, potential overgeneralization or over-averaging of such reports may obviate sector-specific or cluster-specific effects within a society; for instance, the food insecurity and psychosocial implications of COVID-19 among the very poor compared to the well-to-do may not have been well peer-reviewed.

Previous researchers used the structure evaluation and fuzzy TOPSIS methods to assess the security of households and respondents’ perceptions of the socio-economic implications of COVID-19 in Poland and concluded that the ongoing COVID-19 pandemic created a dichotomous deterioration in financial well-being, increased family poverty, led to job losses and increasing unemployment, and led to a situation of an uncertain future among many respondents [9]. In another study, the effect of the COVID-19 pandemic on processes related to higher education was evaluated. The result revealed a mass transition from contact learning to distance learning during the pandemic, and demonstrated evidence of unsustainability of the transition’s impact unless the associated problems of lecturer–student interactions were identified (including the complex problems associated with deterioration in emotional state and reduction in incentives) and solved [10]. In addition, this distance education has disadvantages, including the following: (1) additional time spent to compensate for the distance learning mode; (2) decreased attention paid to the teacher’s words by the audience; (3) increased cost of mistakes in the process of “infinite” communication with students; (4) inaccessible university audience leading to missed students’ opportunities in obtaining knowledge; (5) difficult situations for students wishing to learn; and (6) difficulty of verifying feedback [10,11]. All these factors point to psychological stress on the part of the students and academia as a result of new learning processes.

It should be understood that the coping capacities and strategies to mitigate stressors among the different socio-economic groupings may differ significantly. Whereas some groupings may have fallback mechanisms and reserves in place, others do not. The emergence of COVID-19 globally has caused a large number of economic, political and food system disruptions, especially in low- and middle-income countries (LMICs) (including Nigeria). Access to healthy and sustainable food has remained one of the most debated issues globally in recent times [12]. In 2020–2021, it was revealed that the number of people affected by hunger and food insecurity globally has continued to rise under the influence of the COVID-19 pandemic [13,14]. Furthermore, the prevalence of undernourishment increased globally from 9.3% in 2020 to 9.8% in 2021 under the influence of the COVID-19 pandemic [12]. In addition, Africa remained the region with the heaviest burden of hunger, having 278 million (20.2%) of its population affected by hunger in 2021 [14,15]. In Nigeria, the recent food insecurity situation calls for immediate humanitarian intervention because of a rising number of undernourished people coupled with an estimated 41 percent of the population living in extreme poverty [16,17].

The COVID-19 pandemic led to the institution of several measures to curb the disease’s spread, such as lockdown policies and mobility restrictions, which were associated with a reduction in labor market activities and an increase in food insecurity in Nigeria [18]. Some empirical studies have reported the effect of the COVID-19 pandemic on food security, psychosocial distress, and other socio-economic indicators in Nigeria and globally [19,20,21,22,23,24,25,26,27,28,29]. However, studies on food security and the psychosocial impacts of COVID-19 on Nigerian households are scarce. Therefore, this study aims to assess the impact of COVID-19 on food security and psychosocial stress among Nigerians of different socio-economic groups. The outcomes may assist national authorities to prioritize empirical-based interventions to allocate resources in ways that meet the needs of society and address the current imbalances associated with widening food security and other impacts on society associated with COVID-19. In this cross-sectional survey, we used a questionnaire during field interviews to harvest information from the respondents.

## 2. Materials and Methods

### 2.1. Questionnaire Design

We designed a three-section questionnaire including the following: Section A—questions aimed at collecting general information and the households’ socio-economic and socio-demographic variables, as well as previous histories of illnesses and issues that may bias the outcomes; Section B, which focused on the food security and hunger indices; and Section C, which consisted of questions on psychosocial and stressor information, including the self-perceived quality of life indicators (Appendix A). The questionnaire was designed based on the adapted FAO’s Food Insecurity Experience Scale (FIES) and other validated works [24,30,31,32,33,34]. The developed questionnaire was pretested among five individuals to test for clarity and ease of application without significant assistance. Based on the feedback obtained, the questionnaire was adjusted and distributed among the target population. Informed consent was obtained from each participant before the initiation of the survey and no intrusive questions were asked. Each participant was informed of their right to discontinue the questioning at any stage of the interview.

### 2.2. Field Interview

To reduce the risks of infection and transmission of SARS-CoV-2 (COVID-19) while carrying out the survey, the following precautions were taken by interviewers: (a) a maximum of 10 interviews was carried out per day; (b) disinfectants were utilized liberally; (c) social distancing was observed while conducting the interviews; and (d) other country-specific protocols were observed for mitigating against COVID-19 in Nigeria. We conducted a stratified random sampling of different socio-economic groups based on monthly earning capacity. We used the geo-political stratification of Nigeria to divide the country into the North and the South. Furthermore, in view of the COVID-19 transmission risk and intermittent inter-state movement restrictions at the time of the survey, we selected representative heterogeneous states or territories from each geo-political stratifications, i.e., Abuja (the Federal Capital Territory) and Plateau (141) for the North, and selected Oyo and Lagos (271) for the South (Figure 1). A total of 412 respondents across these locations were sampled.

### 2.3. Data Analysis

All descriptive data were analyzed using proportions (percentages) or the mean with a 95% confidence interval (https://www.openepi.com/Proportion/Proportion.htm, accessed on 17 February 2022). In terms of economic conditions and earning capacity of the respondents, the prevailing earning capacities of the income groupings were classified into three groups as follows: 55,010 NGN (145 USD)/month, 196,280 NGN (516 USD)/month, and 740,375 NGN (USD 1945)/month for the low-, medium-, and high-income earner groups, respectively. We are aware that these earnings may be low compared to those obtain elsewhere. Using the mean values generated for all respondents per income-earning category, the food insecurity experience scale (FIES) was measured using the sliding scale. The COVID-19 acute stress levels were determined using the scale of Van Hoof [33]. Pre- and during-COVID-19 self-rated mean values for stress levels were determined and measured on the scale. Significance of the shift in stress levels was set at an accepted level of α = 0.05 measured using the paired samples *T*-test.

To measure the food security experience of the respondents based on the impact of COVID-19 in Nigeria, an outcome variable (food security and hunger index (FSHI) score) was computed from the total of 24 questions asked on the food security experience scale [35,36,37,38]. The FSHI score ranged from 24 to 293 (maximum obtainable score) with a mean/standard deviation of 133.88 ± 55.05. This score was further categorized as binary (satisfactory or unsatisfactory experience) based on the mean value as the cut-off point [35,36,37,38]. Respondents with scores less or greater than the cut-off point were adjudged to have satisfactory or unsatisfactory experience, respectively, of food security due to the COVID-19 impact. The association between the independent variables (socio-demographic factors) and outcome variable (FSHI score) was determined using the chi-square test and the Fischer’s exact test for 2 × 2 tables. Significant independent variables at *p* < 0.05 were further subjected to a stepwise backward binary logistic regression analysis.

All the self-rated impacts of COVID-19-related experience, stress, and well-being were categorized as binary variables (0 = No or 1 = Yes) and analyzed using the Two by Two Tables in OpenEpi (https://www.openepi.com/TwobyTwo/TwobyTwo.htm, accessed on 14 October 2022). Using a 4-point Likert Scale, the psychological stress and self-rated quality of life of respondents in association with the COVID-19 pandemic for each socio-economic earning group were determined. Finally, the COVID-19 psychosocially non-impacted/slightly impacted persons were evaluated compared to the significantly impacted individuals, and significance of impacts was determined between the two groups. All statistical analyses were conducted using the OpenEpi^®^ software (version 3.01), Atlanta, GA, USA and GraphPad QuickCalcs^®^, San Diego, CA, USA [39,40].

## 3. Results

A total of 412 individuals (typically, the head of the household or his representative) were included, representing 65.78% from the South and 34.22% from the North. Only one person per household was recruited into the survey, thus giving a total of 412 households covered, including 58.15% male-headed and 41.85% female-headed households (male-headed and female-headed mean that a male or female, respectively, makes major decisions and has the breadwinning role in the household). The participants were disaggregated primarily by self-reported income earning capacity per month. We reclassified the income level based on the values provided and mean values per category. Data were also disaggregated by household size, marital status, age categorization, and total hours worked per day (Table 1; Appendix A).

The majority are within the small to medium-sized (1–4) or above medium-sized (5–8) family groupings (93.08%). The large family-sized population (>8 members) accounted for only 6.91% of the total respondents. Furthermore, the largest proportion of the respondents are in the middle age group (21–50 years; 87.59%). A simple majority (53.66%) also work an average of 7–9 h per day. Only 15% of the respondents have previously been hospitalized due to severe illness or surgery, and approximately 10.5% have some form of allergy to medications. Similarly, 10.7% drink alcohol and a further 4.4% smoke. Only 2.9% take recreational drugs but 75.9% drink coffee (Table 1).

In terms of economic conditions and earning capacity of the respondents, the prevalent earning capacities of the income groupings were 55,010 NGN (145 USD)/month, 196,280 NGN (516 USD)/month, and 740,375 NGN (1945 USD)/month for the low-, medium-, and high-income earner groups, respectively. We are aware that these earnings may be low compared to those obtained elsewhere. On average, the low-income earners spend about 19% of their monthly income on transport, compared with approximately 10% for the high-income group. A similar trend exists for the monthly budget for food. A few of the households (4.67%; 95% CI; 3.01–7.18) received assistance from the government in the form of palliatives, financial assistance, and the Nigeria Incentive-Based Risk Sharing System for Agricultural Lending (NIRSAL). Out of the 173 households that ran out of food during the COVID-19 pandemic, 150 (86.71%) reported running out of food more than once. Other sources of financial assistance to augment household needs were sought from cooperatives by 64 respondents (15.76%), while 342 respondents (84.24%) sought help from friends, relatives, and outside cooperative groups (Appendix A).

While the majority of the respondents (households) were low-income earners or in the relatively poor category (74.5%), the high-income earners/upper class were a minority (4.5%). The high-income earners tended to retain food sufficiency with a slight degree of uncertainty regarding the ability to obtain food and a compromise on quality. The middle-income earners tended to remain within a range of uncertainties in their ability to obtain foods with compromised quality and variety, while the low-income earners experienced compromised food quality and variety, and a reduction in food quantity and skipped meals (Figure 2). From the results of FIES, it was found that only high-income households were in the “*food sufficiency*” level (question 1–2), all the low-income households were found between “*compromising on food quality and variety*” *and* “*reducing food quantity and skipping meals*” (question 5–8), while no households were found in the worst level (question 9–10), as indicated in the slider (Figure 2). All categories of households experienced increasing dependency on the general public and a perception of increasing insecurity, with the high-income earners experiencing the greatest shift. In addition, increasing levels of anger and irritation were experienced among all categories (Figure 2). In addition, all categories of income earners experienced a three-point significant shift in the level of acute stress since the beginning of the COVID-19 pandemic (Figure 3). There was a significant difference (*p* = 0.000) in the food security and hunger indices of the respondents before and during COVID-19.

Table 2 presents the socio-demographic factors associated with food security and the hunger experience of the respondents due to the impact of COVID-19 in Nigeria. Of the socio-demographic variables, only gender (*p* = 0.012), level of education of household head (*p* = 0.000), work hours per day (*p* = 0.006), and family income based on society class (*p* = 0.000) were significantly associated with food security and hunger due to the COVID-19 pandemic. The results of the regression showed that females were 0.59 times (95% CI: 0.40–0.88, *p* = 0.013) less likely to have a satisfactory experience of food security due to the pandemic than males. Household heads with at least a diploma level of education were more likely to have a satisfactory experience with regards to food security due to the pandemic. Respondents with postgraduate degrees were at least nine times (95% CI: 3.16–28.74, *p* < 0.001) more likely to have a satisfactory experience of food security and hunger due to the pandemic than household heads with at most a primary level of education. Participants with >12 work hours per day were also more likely (OR: 3.35, 95% CI: 1.29–8.66, *p* = 0.019) to demonstrate a satisfactory experience of food security than those working 2–6 h daily. Lastly, those with medium (OR: 4.36, 95% CI: 2.64–7.21, *p* < 0.001) and high (OR: 30.27, 95% CI: 3.99–229.90, *p* < 0.001) family income based on society class were more likely to have a satisfactory experience of food security and hunger than respondents with low family income.

Further, irrespective of the social class of the respondents, less than a quarter of the participants had negative experiences, such as movement/change in location within the same city (17.68%), beginning a new relationship (14.43%), recent change in job/loss of job (11.55%), movement to another city (10.81%), or separation from spouse or long-term relationship (10.07%) (Table 3). A few respondents experienced deaths of a family member (4.91%) and legal problems (4.18%). However, almost half (43.38%) of the respondents experienced financial difficulties (Table 3). Regarding well-being, a few of the respondents felt more under pressure at work (25.91%), lived by themselves (20.44%), felt lonely (14.91%), felt under pressure during the day (13.64%), had serious arguments with close relatives (10.06%), had more problems with colleagues at work (10.05%), or were unable to find a job (9.58%) due to the COVID-19 pandemic. However, 231 (60.16%) reported satisfaction with their jobs despite the pandemic (Table 3).

Of the total respondents, following the advent of COVID-19, a total of 4 (1.0%) felt awful/terrible, 41 (10.2%) felt poor, 152 (37.7 %) felt fair, 145 (36.0%) felt good, and 61 (15.1 %) were reported to have excellent feelings (Table 4). Though the majority of respondents were low- and middle-income earners, there were significant differences among the coping capacities of the respondents of different income earning groups (Table 4). Generally, among all variables investigated, respondents experienced lower psychological stress due to the impact of COVID-19. For instance, very low proportions of respondents experienced restless nights, feeling dizzy or fainting, irritability, sadness or depression, panic attacks, a perception of having a wrongly diagnosed physical COVID-19-related problem, or, after reading or hearing about COVID-19, a feeling of having similar symptoms (Table 4).

By comparison, among the categories of earned income, the psychological stress due to the impact of COVID-19 was observed to be more in the low-income earning group than in other groups. Out of 74 responses, 58 (78.38%) from the low-income group reported experiencing a long time to fall asleep. The low-income earners also reported having to significantly deal with restless sleep or nights, compared to other groups, i.e., medium- and high-income categories. Similarly, palpitations (62.50%), feeling dizzy/like fainting (77.27%), tiredness or lack of energy (80.05%), sadness or depression (75.61%), feeling tense (83.05%), loss of interest in things (76.19%) and panic attacks (85.71%) were more frequent among the low-income earners than the medium- or high-income earners (Table 4).

## 4. Discussion

We evaluated the food security impacts and the psychosocial and economic implications of COVID-19 on different income-earning groups in Nigeria, an example of a lower-middle-income economy, and presented our findings. We classified the respondents based on different disaggregated criteria (marital status, household size, age, gender, education, hospitalization, routine behaviours, and income-earning capacity), factors that may affect the perception and responses to the subject of this study, and key highlights were presented in Table 1 and Appendix A. However, with the understanding that the monthly income-earning capacity is a major influence in determining food security in urban and peri-urban households, we used income-earning capacity as a basis for further evaluation. This factor also tends to affect the locational clustering of respondents, health, well-being, and coping capacities in response to health challenges—in this case, the COVID-19 pandemic [41]. This appears to be one of the first peer-reviewed evaluations of how different income earning groups are impacted differently by COVID-19, especially in Nigeria.

There was a wide disparity in average income-earning capacity per economic group in our assessment (USD 145–1945), and, consequently, the ability of the infected patient or directly affected families to respond to critical health situations. The universal health coverage in Nigeria is still underdeveloped, with the country having a lower UHC (1.1%) compared with countries such as Ghana (49.1%) and Kenya (18.2%) [42,43,44]. This is an indication that high out-of-pocket expenses for healthcare are still prevalent. Furthermore, less than 5% of the population, which is greater than 200 million, can afford health care provided through private insurance [42,43]. Furthermore, the National Health Insurance Scheme (NHIS) of Nigeria’s Federal Ministry of Health targets only government employers, which leaves the majority of Nigerians without appropriate health cover. Consequently, if an overwhelming illness arises because of COVID-19, low-income earners and possibly middle-income earners are unlikely to be able to pay for the cost associated with hospitalization, and may completely avoid seeking treatment in hospitals, possibly with more fatal consequences [45]. In Kenya, per-day, per-patient unit costs for asymptomatic patients and patients with mild-to-moderate COVID-19 disease receiving home-based care range between USD 18.89 and 18.99, respectively [46]. However, in an isolation center or hospital, the same unit costs for asymptomatic patients and patients with mild-to-moderate disease are USD 63.68 and 63.70, respectively, and for critical cases with possible admission to intensive care units, they may increase to between USD 124.53 and 599.51 per day per patient [46]. In the USA, the median charge for hospitalization of a COVID-19 patient over a course of treatment until discharged ranged from USD 34,662 for the 23–30 age group to USD 45,683 for the 51–60 age group [47,48]. Although no peer-reviewed study estimating the cost of hospitalization for a COVID patient in Nigeria currently exists, anecdotal estimation places the average cost at between USD 750 and 13,000 per person, depending on the duration of hospitalization [49]. These observations have some implications: there may be some distortion of the national epidemiological (morbidity–mortality) data and related health costs associated with COVID-19 in Nigeria, because individuals in the low- and middle-income groups would most likely shun hospitalization and post mortem examinations to determine the cause of death largely due to the associated huge costs [50]. To date, Nigeria has only reported 253,875 confirmed cases with 3139 human deaths [4].

Generally, there was a significant difference (*p* = 0.000) in the food security and hunger indices of the respondents before and during COVID-19. In addition, based on the FIES per economic group, the group with the worst capacity to respond to and pay for hospitalization associated with COVID-19 (i.e., the low-income earners) is the same group that has the worst experience according to the FIES (ranging from compromised food quality and variety, to reduced food quantities and skipping meals). This is an indication that, in addition to a worsening health situation due to COVID-19 and inability to pay for medication, such families may also experience food insecurity, hunger, and deprivation. Previous works have reached similar conclusions, namely, that the COVID-19 pandemic has affected various dimensions of food security and households’ incomes in developing countries, and among low-income households in high-income-economies such as the US [47,48,49]. Furthermore, Olwande et al. [50] showed that the COVID pandemic caused a significant decline in households’ incomes and had negative social and economic impacts on households living in both the urban and rural areas of Kenya. Consequently, households experienced less food consumption and reduced food quality. In addition, Balana et al. [24] found that income losses due to the devastating effect of COVID-19 have pushed more households in Nigeria into a more severe food insecurity status. In addition, the FIES (Figure 1) showed that low-income households were the most affected income category in terms of food insecurity experiences amid the COVID-19 pandemic in the study areas in Nigeria. This result was corroborated by Balana et al. [24], who found that income losses pushed Nigerian households further into a more severe food insecurity level, which is indicative of the challenges of achieving zero hunger (SDG 2) in Nigeria by 2030 [51,52]. It is recommended that governments, particularly in the low- and lower-middle-income countries, should consider robust food supports and palliatives targeted at the poor and low-income earners, since these categories of individuals have experienced significantly reduced and compromised food quantity and quality [53,54,55].

In this study, gender (*p* = 0.012), level of education of household head (*p* = 0.000), work hours per day (*p* = 0.006), and family income based on society class (*p* = 0.000) were significantly associated with food security and hunger due to the COVID-19 pandemic. This agrees with the findings from the FIES, and confirms that the poor, females, and the uneducated are more desirous of government assistance during pandemics such as COVID-19 than the more affluent, male, and formally educated individuals. Unfortunately, these categories may experience denied service delivery during such food distributions [56]. Furthermore, respondents working greater than 12 h per day were more likely to be highly skilled, and those with higher family incomes demonstrated a more satisfactory experience of food security than those working 2–6 h daily. This indicates that lower priority should be given to this category in situations of assistance associated with pandemics.

With regards to psychosocial stress and mental well-being, all categories of income earners experienced some significant shift (*p* < 0.001) in acute stress associated with the ongoing COVID-19 pandemic (Figure 2). Van Hoof [33] and other workers associated COVID-19 with a secondary epidemic of burnouts and stress-related absenteeism, low mood, insomnia, stress, anxiety, anger, irritability, emotional exhaustion, depression, and post-traumatic stress symptoms [57,58,59]. Similarly, in China, a meta-analysis investigation into the mental health impact of the COVID-19 epidemic on the general population highlighted the pooled prevalence of high stress, followed by depression and anxiety [60]. In Nigeria, the issue of mental health and related research are poorly considered, both by the authorities and society, and mental health policies and mental health infrastructures are still underdeveloped [61]. There is therefore a need to further explore, on a larger scale, how the COVID-19 pandemic has impacted the mental health of Nigerians.

### Study Limitations

Our study is subject to certain limitations. The sample size is small, particularly for the high-income group. Although the sampling was stratified and randomized, we did not achieve a significantly large number of high-income earners. This observation may be a true reflection of the population dynamics in Nigeria. Secondly, only few states were selected to represent the South and the North. It should be understood that under the situation of the pandemic, government-imposed restrictions applied during the period of the assessment, and this interrupted or limited movements. In addition, the sporadic insecurity (especially due to banditry and kidnapping on highways) complicated access to some locations and reduced interconnections between many cities during the survey. Perhaps a cross-country multi-regional survey with a much larger sample size would yield a different outcome if applied in future studies. We are aware that using other excellent analytic tools such as the fuzzy clustering method (e.g., fuzzy TOPSIS) may better describe the present level of the threat and show whether and to what extent the problem should be addressed, while at the same time predicting useful tools to address the problem [9,62]. We recommend the application of this method in future studies.

As was evident in other geographies, the COVID-19 pandemic significantly affected households in Nigeria in terms of food and socio-economic security. Kalinowski et al. [9] previously confirmed that the COVID-19 pandemic increased the level of uncertainty about the future situation and the poverty of households in Poland; the authors suggested that the impacts of government responses (public policies introduced to combat COVID-19) were positive but insufficient in terms of their effect on households, and that there was a need to increase the effectiveness of these measures. Our work agreed with this position and we support the recommendation. We observed that lockdowns and restrictions hindered businesses with negative consequences for food insecurity. A similar conclusion was previously reached [9]. It was recommended that such restrictions must be accompanied by adequate social programs for societal acceptability [9]. The utilization of empirical assessments such as the one in this study to shape effective policies might mitigate the consequences of the COVID-19 pandemic while, at the same time, increasing social comfort.

## 5. Conclusions

In the present study, we elucidated the mental health issues experienced by Nigerians based on their income group. Although all income earners experienced certain changes in their mental health, the low- and sometimes the middle-income earners reported the worst experience. These incomes categories experienced more pressure because of irrational behavior, sadness or depression, restlessness, panic attacks, and loss of motivation, and may have the worst state of well-being compared to more affluent individuals and high-income earners in society [63]. We are aware that patients may move from panic attacks into denial, a situation that may complicate responses to COVID-19 [64].

Based on our findings, it is recommended that: (1) full mapping of the affected households should be conducted to determine the full severity and impact of the pandemic on health, social, economic, and mental wellness; (2) the government should show more political commitments and transparency in the execution of social and economic policies and deliveries, particularly targeted at the poor and low-income earners, whose food security and psychology were more impacted by the COVID-19 pandemic; (3) specific household needs and mitigation plans should be disaggregated per income group so as to meet the specific societal needs; and (4) targeted counselling sessions and centers should be established to cater for psychosocially impacted and stressed individuals.

## Figures and Tables

**Figure 1 ijerph-20-04016-f001:**
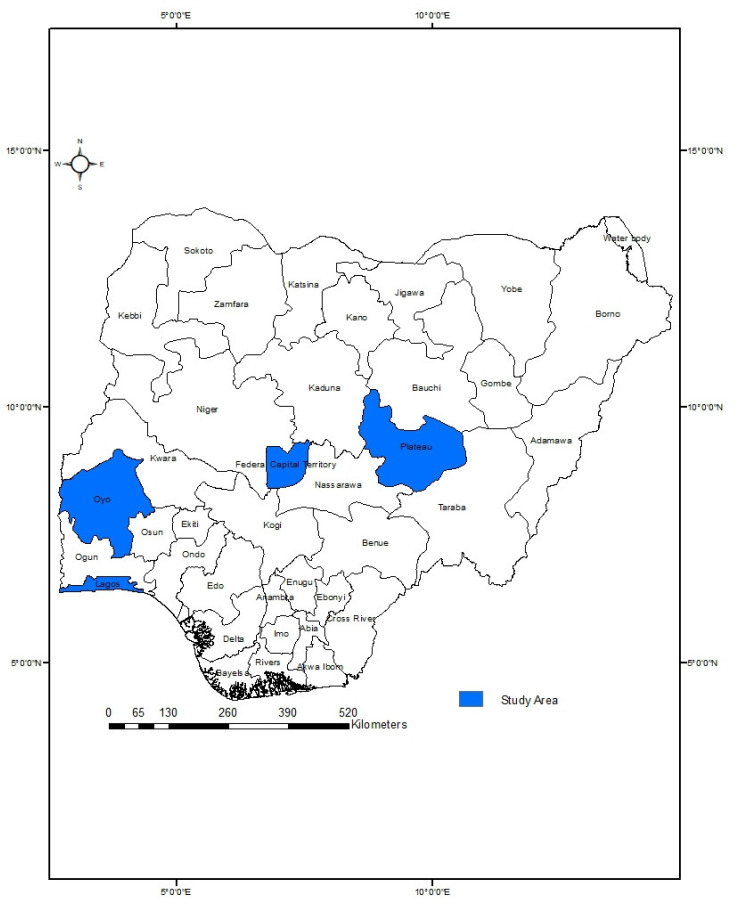
Map of Nigeria showing the study area.

**Figure 2 ijerph-20-04016-f002:**
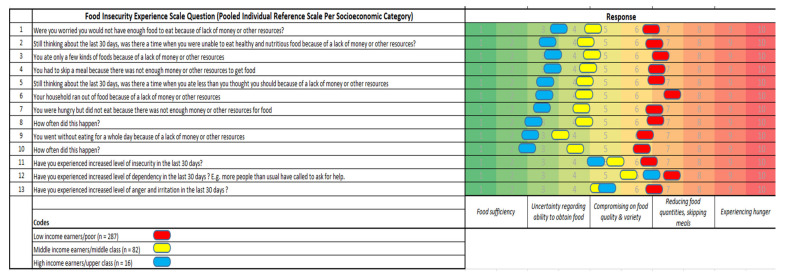
Modified food insecurity experience scale (FIES) pooled per socio-economic category during the 2020–2021 COVID-19 pandemic, Nigeria.

**Figure 3 ijerph-20-04016-f003:**
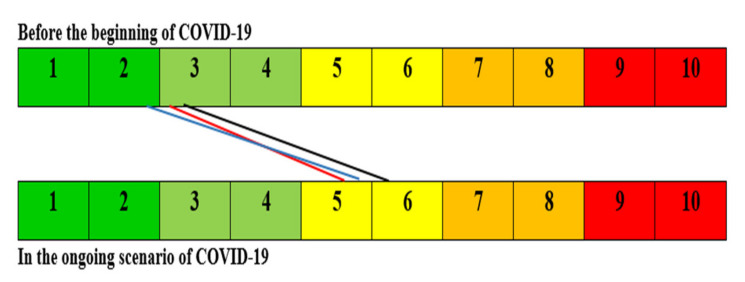
Self-reported levels of acute stress among respondents per socio-economic earning category during the ongoing COVID-19 pandemic, Nigeria. (Note that: 1 = no stress and 10 = much stressed [33]. Black line represents the low-income earners, red line the middle-income earners, and blue line the high-income earners. There was significant acute stress observed in each category: low-income earners (n = 289) (slide from 3.28 ± 1.68 to 6.21 ± 2.03, *p* value < 0.0001); middle-income earners (n = 87) (slide from 3.18 ± 1.88 to 5.53 ± 2.47, *p* value < 0.0001); high-income earners (n = 16) (slide from 2.89 ± 1.41 to 5.72 ± 2.52, *p* value < 0.001)).

**Table 1 ijerph-20-04016-t001:** Descriptive data of respondents regarding the psychosocial and food security-related impacts of COVID-19, Nigeria.

Variable (*n*)	Classification	Number	%
Marital status * (411)	Single	132	32.12
	Married	255	62.04
	Separated/Divorced	6	1.46
	Widowed	18	4.38
Total number of persons in the household (405)	1–4	190	46.91
	5–8	187	46.17
	9–12	23	5.68
	> 12	5	1.23
Age (411)	≤20 years	7	1.70
	21–30 years	105	25.55
	31–40 years	151	36.74
	41–50 years	104	25.30
	>50 years	44	10.71
Gender (411)	Male	239	58.15
	Female	172	41.85
Level of education of household head (358)	≤primary	25	6.99
	Secondary	69	19.27
	Diploma—first degree	193	53.91
	MSc and PhD	71	19.83
Work hours per day (382)	2–6 h	24	6.28
	7–9 h	205	53.66
	10–12 h	62	16.23
	>12 h	91	23.82
∞ Family income based on society class (411)	Low	295	71.78
	Medium	97	23.60
	High	19	4.62
Previously hospitalized for severe illness/surgery (405)	Yes ^#^	61	15.06
	No	344	84.94
Allergic to medication ** (401)	Yes	42	10.47
	No	359	89.53
Drink alcohol routinely or periodically (410)	Yes	44	10.73
	No	366	89.27
Smoke routinely or periodically (410)	Yes	18	4.39
	No	392	95.61
Take recreational drugs routinely or periodically (410)	Yes	12	2.93
	No	398	97.07
Drink coffee and tea regularly (410)	Yes	311	75.85
	No	99	24.15

* One person (0.25%) did not state their marital status clearly. ∞ Family incomes—N55,010 (US$145)/month; N196,280 (US$516)/month and N740,375 (US$1945)/month for the low, medium and high-income earner groups respectively. ^#^ Only 22/61 (36.0%) persons reported abortion, miscarriage, bronchitis, appendectomy, blood pressure, surgery, eye defects, hernia, rheumatism, cataract, nephritis, and diabetes. ** Only six (6) individuals declared that they were on verifiable chronic medication during the period of the survey. Of the 412 respondents, 271 (65.78%) were from the South (Oyo and Lagos) and 141 (34.22%) from the North (Plateau and Abuja). A total of 201/382 (52.62%) of the spouses of respondents were gainfully employed in other types of jobs.

**Table 2 ijerph-20-04016-t002:** Socio-demographic factors associated with food security and hunger experience of the respondents due to the impact of COVID-19 in Nigeria.

Variable (*n*)	Classification	Number (%)	Unsatisfactory Experience	Satisfactory Experience	*p* Value (χ^2^)	OR	95% CI	*p* Value
Marital status * (412)	Single	132 (32.04)	72	60	0.669	-	-	-
	Married	255 (61.89)	127	128		-	-	-
	Separated/Divorced	6 (1.46)	4	2		-	-	-
	Widowed	18 (4.37)	10	8		-	-	-
Total number of persons in the household (405)	1–4	190 (46.91)	107	83	0.158	-	-	-
	5–8	187 (46.17)	90	97		-	-	-
	9–12	23 (5.68)	10	13		-	-	-
	>12	5 (1.23)	1	4		-	-	-
Age (411)	≤20 years	7 (1.70)	4	3	0.991	-	-	-
	21–30 years	105 (25.55)	56	49		-	-	-
	31–40 years	151 (36.74)	77	74		-	-	-
	41–50 years	104 (25.30)	54	50		-	-	-
	>50 years	44 (10.71)	22	22		-	-	-
Gender (411)	Male	239 (58.15)	111	128	0.012 ^α^	1.00	-	-
	Female	172 (41.85)	102	70		0.59	0.40, 0.88	0.013 ^α^
Level of education of household head (358)	≤primary	25 (6.99)	20	5	0.000 ^α^	1.00	-	-
	Secondary	69 (19.27)	54	15		1.11	0.36, 3.46	>0.999
	Diploma–first degree	193 (53.91)	85	108		5.08	1.83, 14.1	0.001 ^α^
	MSc and PhD	71 (19.83)	21	50		9.52	3.16, 28.74	<0.001 ^α^
Work hours per day (382)	2–6 h	24 (6.28)	16	8	0.006 ^α^	1.00	-	-
	7–9 h	205 (53.66)	112	93		1.66	0.68, 4.05	0.366
	10–12 h	62 (16.23)	38	24		1.26	0.47, 3.40	0.838
	>12 h	91 (23.82)	34	57		3.35	1.29, 8.66	0.019 ^α^
∞ Family income based on society class (411)	Low	295 (71.78)	185	110	0.000 ^α^	1.00	-	-
	Medium	97 (23.60)	27	70		4.36	2.64, 7.21	<0.001 ^α^
	High	19 (4.62)	1	18		30.27	3.99, 229.90	<0.001 ^α^
Previously hospitalized for severe illness/surgery (405)	Yes #	61 (15.06)	179	165	0.579	-	-	-
	No	344 (84.94)	29	32		-	-	-
Allergic to medication ** (401)	Yes	42 (10.47)	25	17	0.330	-	-	-
	No	359 (89.53)	183	176		-	-	-
Drink alcohol routinely or periodically (410)	Yes	44 (10.73)	26	18	0.341	-	-	-
	No	366 (89.27)	187	179		-	-	-
Smoke routinely or periodically (410)	Yes	18 (4.39)	9	9	1.000	-	-	-
	No	392 (95.61)	204	188		-	-	-
Take recreational drugs routinely or periodically (410)	Yes	12 (2.93)	8	4	0.385	-	-	-
	No	398 (97.07)	205	193		-	-	-
Drink coffee and tea regularly (410)	Yes	311 (75.85)	154	157	0.084	-	-	-
	No	99 (24.15)	59	40		-	-	-

* One person (0.25%) did not state their marital status clearly. OR—Odds ratio; CI—Confidence interval; α—Significant at *p* < 0.05; ∞—55,010 NGN (145 USD)/month, 196,280 NGN (516 USD)/month, and 740,375 NGN (1945 USD)/month for the low-, medium-, and high-income earner groups, respectively. # Only 22/61 (36.0%) persons reported abortion, miscarriage, bronchitis, appendectomy, blood pressure, surgery, eye defects, hernia, rheumatism, cataract, nephritis, and diabetes. ** Only six (6) individuals declared that they were on verifiable chronic medication during the period of the survey.

**Table 3 ijerph-20-04016-t003:** Self-rated impact of COVID-19-related selected experience and well-being of respondents.

Variable (*n*)	Yes		No	
Selected Experience	Number	%	Number	%
Death of a family member (407)	20	4.91	387	95.09
Separation from spouse or long-term relationship (407)	41	10.07	366	89.93
Recent change in job/loss of job (407)	47	11.55	360	88.45
Financial difficulties (408)	177	43.38	231	56.62
Movement/change in location within the same city (311)	55	17.68	256	82.32
Movement to another city (407))	44	10.81	363	89.19
Legal problem (407)	17	4.18	390	95.82
Begin a new relationship (409)	59	14.43	350	85.57
**Well-Being**				
Satisfaction with job (384)	231	60.16	153	39.84
Felt more under pressure at work (382)	99	25.91	283	74.08
Have more problem with colleagues at work (378)	38	10.05	340	89.95
Retired person or student (349)	49	14.04	300	85.96
Felt under pressure during the day (352)	48	13.64	304	86.36
Unable to find a job due to COVID-19 (355)	34	9.58	321	90.42
Have serious arguments with close relatives (358)	36	10.06	322	89.94
Have serious arguments with other people (261)	23	8.81	238	91.19
Close relatives been seriously ill due to COVID-19 (363)	27	7.44	336	92.56
Felt tension at home (365)	61	16.71	304	83.29
Live by oneself (367)	75	20.44	292	79.56
Felt lonely (369)	55	14.91	314	85.09

**Table 4 ijerph-20-04016-t004:** Psychological stress and self-rated quality of life of respondents per socio-economic earning category due to the impact of COVID-19.

Variable (*n*)	Income Category	Not at All (%)	Only a Little (%)	Somewhat Much (%)	A Great Deal (%)
It takes a long time to fall asleep	Low (294)	159 (54.08)	77 (26.19)	47 (15.99)	11 (3.74)
	Medium (96)	62 (64.58)	21 (21.88)	9 (9.38)	4 (4.17)
	High (19)	12 (63.16)	4 (21.05)	3 (15.79)	0 (0.00)
Restless sleep	Low (294)	152 (51.70)	77 (26.19)	53 (18.03)	12 (4.08)
	Medium (96)	64 (66.67)	26 (27.08)	3 (3.13)	3 (3.13)
	High (19)	16 (84.21)	2 (10.53)	0 (0.00)	1 (5.26)
Waking too early and not being able to fall asleep again	Low (294)	146 (49.66)	83 (28.23)	42 (14.29)	23 (7.82)
	Medium (96)	54 (56.25)	27 (28.13)	7 (7.29)	8 (8.33)
	High (19)	15 (78.95)	1 (5.26)	1 (5.26)	2 (10.53)
Feeling tired on waking up	Low (294)	168 (57.14)	81 (27.55)	33 (11.22)	12 (4.08)
	Medium (96)	45 (46.88)	40 (41.67)	8 (8.33)	3 (3.13)
	High (19)	16 (84.21)	0 (0.00)	2 (10.53)	1 (5.26)
Chest, stomach, or abdominal pain	Low (294)	220 (74.83)	54 (18.37)	17 (5.78)	3 (1.02)
	Medium (96)	81 (84.38)	11 (11.46)	3 (3.13)	1 (1.04)
	High (19)	18 (94.74)	1 (5.26)	0 (0.00)	0 (0.00)
Heart beating quickly or strongly (palpitation) without a reason like exercise	Low (294)	243 (82.65)	36 (12.24)	13 (4.42)	2 (0.68)
	Medium (96)	78 (81.25)	10 (10.42)	5 (5.21)	3 (3.13)
	High (19)	15 (78.95)	3 (15.79)	0 (0.00)	1 (5.26)
Feeling dizzy or like fainting	Low (293)	240 (81.91)	36 (12.29)	16 (5.46)	1 (0.34)
	Medium (95)	86 (90.53)	6 (6.32)	2 (2.11)	1 (1.05)
	High (19)	15 (78.95)	2 (10.53)	2 (10.53)	0 (0.00)
Feeling pressure or tightness in the head or body	Low (292)	215 (73.63)	61 (20.89)	13 (4.45)	3 (1.03)
	Medium (96)	75 (78.13)	16 (16.67)	4 (4.17)	1 (1.04)
	High (19)	16 (84.21)	1 (5.26)	0 (0.00)	2 (10.53)
Breathing difficulties or feeling of not having enough air	Low (292)	243 (83.22)	29 (9.93)	17 (5.82)	3 (1.03)
	Medium (96)	83 (86.65)	8 (8.33)	5 (5.21)	0 (0.00)
	High (19)	16 (84.21)	2 (10.53)	0 (0.00)	1 (5.26)
Feeling tired or lack of energy	Low (294)	190 (64.63)	55 (18.71)	28 (9.52)	21 (7.14)
	Medium (96)	61 (63.54)	26 (27.08)	7 (7.29)	2 (2.08)
	High (19)	16 (84.21)	2 (10.53)	1 (5.26)	0 (0.00)
Irritable	Low (294)	192 (65.31)	73 (24.83)	24 (8.16)	5 (1.70)
	Medium (96)	73 (76.04)	16 (16.67)	7 (7.29)	0 (0.00)
	High (19)	14 (73.70)	4 (21.05)	0 (0.00)	1 (5.26)
Sad or depressed	Low (293)	175 (59.73)	87 (29.69)	21 (7.17)	10 (3.41)
	Medium (96)	65 (67.71)	23 (23.96)	5 (5.21)	3 (3.13)
	High (19)	14 (73.70)	3 (15.79)	2 (10.52)	0 (0.00)
Feeling tensed or ‘wound up’	Low (293)	188 (63.95)	56 (19.05)	41 (13.95)	8 (13.95)
	Medium (97)	67 (69.07)	21 (21.65)	7 (7.22)	2 (2.06)
	High (19)	16 (84.21)	2 (10.52)	0 (0.00)	1 (5.26)
Lost interest in most things	Low (293)	201 (68.60)	60 (20.48)	22 (7.51)	10 (3.41)
	Medium (97)	67 (69.07)	21 (21.65)	6 (6.19)	3 (3.09)
	High (19)	16 (84.21)	2 (10.52)	1 (5.26)	0 (0.00)
Attack or panic	Low (294)	232 (78.91)	38 (12.92)	22 (7.48)	2 (0.68)
	Medium (97)	85 (87.63)	8 (8.25)	4 (4.12)	0 (0.00)
	High (19)	18 (94.73)	1 (5.26)	0 (0.00)	0 (0.00)
Perception of having a physical COVID-19 related problem wrongly diagnosed	Low (293)	256 (87.37)	27 (9.21)	9 (3.07)	1 (0.34)
	Medium (97)	88 (90.72)	7 (7.22)	2 (2.06)	0 (0.00)
	High (19)	19 (100.00)	0 (0.00)	0 (0.00)	0 (0.00)
After reading or hearing about COVID-19, feeling of having similar symptoms	Low (293)	251 (85.67)	22 (7.50)	19 (6.48)	1 (0.34)
	Medium (97)	83 (85.57)	10 (10.31)	2 (2.06)	2 (2.06)
	High (19)	18 (94.73)	1 (5.26)	0 (0.00)	0 (0.00)
When I noticed a sensation in your nose, nostrils, trachea, or chest, or I coughed, I find it difficult to think of something else	Low (293)	240 (81.91)	31(10.58)	13 (4.44)	9 (3.07)
	Medium (97)	75 (77.32)	13 (13.40)	4 (4.12)	5 (5.15)
	High (19)	15 (78.95)	2 (10.52)	1 (5.26)	1 (5.26)

Of the total 403 respondents, following the advent of COVID-19, a total of 4 (1.0 %) felt awful/terrible, 41 (10.2 %) felt poor, 152 (37.7 %) felt fair, 145 (36.0 %) felt good, and 61 (15.1 %) reported to be have excellent feelings.

## Data Availability

The data used in this study are available upon reasonable request from the corresponding author.

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
