# Peer review of "COVID-19 Pandemic Impacted Food Security and Caused Psychosocial Stress in Selected States of Nigeria"

_ijerph, 2023, doi:10.3390/ijerph20054016_

Round 1
Reviewer 1 Report
The manuscript entitled "COVID-19 pandemic impacts food security and causes psychosocial stress in Nigeria" (ijerph-2186142) was aimed at studying the effects of the COVID-19 pandemic on the level of food security in Nigeria with a focus on socio-psychological implications.
(1) Please consider redesigning your abstract, as now it is filled with in-depth with empirical results (for example the p-values of socio-demographic variables; odds ratios and confidence intervals). The journal proposes an abstract of about 200 words (yours is almost double). The abstract should include information on: (a) the background – a broad presentation of the context and highlight the research objective based on literature gap; (b) methods – briefly describe the main methodological procedures; (c) results – summarize(!) the article’s main findings; (d) conclusions, practical and managerial implications.
(2) The formatting of the manuscript needs adjustments. Please correct the landscape format where it is not needed.
(3) The research objective is presented in-between lines 59-64 and in relation with the literature gap. However, I consider it would be beneficial to specifically include the notion of "research objective" or "research aim" in this part of the paper.
(4) The Introduction section should end with a paragraph dedicated to the presentation of the manuscript structure.
(5) Line 73: I could not find Appendix 1 in the manuscript, nor in the supplementary files.
(6) There is a lack of explanations regarding the construction of the food security and hunger indices score in Section 2. What is in the composition of this outcome variable? Why did you resort to 4-point Likert Scale instead of the tradition 5-point? (Line 114).
(7) I suggest dividing the last column from Table 1 into two: one for number and one for percentage. Please pay attention to the sum of percentages, because they do not always result in 100%. For example, for the total number of persons in the household: 46.91% + 46.17% + 5.68% + 1.23% = 99.99%. Family income based on society class needs cell merging. Why are the totals so divergent from 412 individuals, as stated in line 120? For the marital status I noticed the explanation, but for the rest? Also applies for Table 3 and 4.
(8) Please improve the quality of Figures 2 and 3. Figure 2 is a screenshot, but I believe that looks like it was generated in Excel. Please consider pasting the results as a table and format it properly in the manuscript. Some of the 13 "questions" from Figure 2 are formulated as affirmations instead of questions. Line 172: format the reference according the MDPI style.
(9) Line 256: I suggest changing "the first" to "one of the first peer-reviewed evaluations".
(10) I suggest moving the presentation of the study limitations (lines 339-350) to the Conclusions section.
(11) Please consider elaborating one more paragraph in the Conclusions section for touching on possible future research avenues based on the current limitations.
Author Response
We Thank the reviewer for the insightful and useful suggestions. We have revised the document and addressed the comments as found below:
(1) Please consider redesigning your abstract, as now it is filled with in-depth with empirical results (for example the p-values of socio-demographic variables; odds ratios and confidence intervals). The journal proposes an abstract of about 200 words (yours is almost double). The abstract should include information on: (a) the background – a broad presentation of the context and highlight the research objective based on literature gap; (b) methods – briefly describe the main methodological procedures; (c) results – summarize(!) the article’s main findings; (d) conclusions, practical and managerial implications.
Response: This is now fixed and the statistics reduced in favor of the narratives.
(2) The formatting of the manuscript needs adjustments. Please correct the landscape format where it is not needed.
The formatting issues of the manuscript has been fixed.
(3) The research objective is presented in-between lines 59-64 and in relation with the literature gap. However, I consider it would be beneficial to specifically include the notion of "research objective" or "research aim" in this part of the paper.
We have edited the manuscript to bring out, specifically, the aim of the aim.
(4) The Introduction section should end with a paragraph dedicated to the presentation of the manuscript structure.
We have included a statement showing the main direction of the study.
(5) Line 73: I could not find Appendix 1 in the manuscript, nor in the supplementary files.
Appendix 1 is now attached.
(6) There is a lack of explanations regarding the construction of the food security and hunger indices score in Section 2. What is in the composition of this outcome variable? Why did you resort to 4-point Likert Scale instead of the tradition 5-point? (Line 114).
We thank the reviewer for these comments. However, we already explained in detail the construction of the outcome variable, Food Security and hunger indices (FSHI) score, in lines 133-145 under the data analysis section 2.3.
The use of a 4-point Likert Scale for the psychological stress and self-rated quality of life of respondents was as provided in the questionnaire designed based on the adapted FAO’s Food Insecurity Experience Scale (FIES).
(7) I suggest dividing the last column from Table 1 into two: one for number and one for percentage. Please pay attention to the sum of percentages, because they do not always result in 100%. For example, for the total number of persons in the household: 46.91% + 46.17% + 5.68% + 1.23% = 99.99%. Family income based on society class needs cell merging. Why are the totals so divergent from 412 individuals, as stated in line 120? For the marital status I noticed the explanation, but for the rest? Also applies for Table 3 and 4.
We have added another column to the tables 1 and 3 to accommodate splitting the numbers from percentages. We don’t think it is necessary to include this for table 2 since with have other columns for the satisfactory and unsatisfactory responses. Numbers may not always add up to 100% because of rounding off of numbers or where missing data exist.
(8) Please improve the quality of Figures 2 and 3. Figure 2 is a screenshot, but I believe that looks like it was generated in Excel. Please consider pasting the results as a table and format it properly in the manuscript. Some of the 13 "questions" from Figure 2 are formulated as affirmations instead of questions. Line 172: format the reference according the MDPI style.
The original files are now attached. The picture qualities in the original submissions were good enough but faded in the reviewed manuscript.
(9) Line 256: I suggest changing "the first" to "one of the first peer-reviewed evaluations".
We have changed this phrase.
(10) I suggest moving the presentation of the study limitations (lines 339-350) to the Conclusions section.
We thank the reviewer for this suggestion. However, we plan to retain the study limitation in its current location. We believe presentation of study limitations always come at the end of the discussion just before the conclusion.
(11) Please consider elaborating one more paragraph in the Conclusions section for touching on possible future research avenues based on the current limitations.
We thank the reviewer for these. We have captured this in lines 406-407.

Reviewer 2 Report
This is a manuscript review for ''COVID-19 pandemic impacts food security and causes psychosocial stress in Nigeria''
The abstract is too long. Reduce to about 200 words as per the guide.
Line 53: Change ' time' with 'times'
Line 81: Replace 'taking' with ' taken'
Field interview: Four states are not representative of Nigeria (36 states +FCT). Abuja and Plateau are not real north. The title should be changed to
COVID-19 pandemic impacts food security and causes psychosocial stress in some states of Nigeria
Line 104: Provide a reference for FISH score
Line 122: revise English. Explain female and male-headed
Line 290: Put the p-value after 'significance'
Line 323; use 'greater than instead of >
Author Response
We thank the reviewer for a good review.
We addressed the comments as found below:
This is a manuscript review for ''COVID-19 pandemic impacts food security and causes psychosocial stress in Nigeria''
The abstract is too long. Reduce to about 200 words as per the guide.
Line 53: Change ' time' with 'times'
This has been changed.
Line 81: Replace 'taking' with ' taken'
This has been replaced.
Field interview: Four states are not representative of Nigeria (36 states +FCT). Abuja and Plateau are not real north. The title should be changed to
COVID-19 pandemic impacts food security and causes psychosocial stress in some states of Nigeria
The title of the manuscript has been modified accordingly.
Line 104: Provide a reference for FISH score
We have included references for the score computation.
Line 122: revise English. Explain female and male-headed
We have added explanation of these words ‘(male-headed and female-headed mean that major decisions and breadwinning role in the households are made by the male and female respectively)’.
Line 290: Put the p-value after 'significance'
We have included the p value.
Line 323; use 'greater than instead of >
We have used greater than in place of >.

Reviewer 3 Report
The article, although it raises important issues of the day, has a number of errors which, at this point, I believe make it difficult to publish. As this is an important issue it should be widely recognised, and its main aspects included.
Comments:
1) There is a lack of a thorough review of the literature; the problem has been addressed many times in the world, which the authors have not indicated.
2) It would be useful to show how one moved from panic to denial of the problem. I myself had the opportunity to describe this phenomenon.
3) It is worth noting that, using a fuzzy clustering method, it is possible to describe the threat states themselves, and to show whether and to what extent the problem should be addressed, but also which tools should be used.
(4) In the introduction, apart from the lack of key literature on the problem studied, the aim of the paper, the research questions or the hypothesis are missing. It is difficult to analyse an article when we do not know what the author wants to show.
5) I am very unsatisfied with the explanation of the research method and recommendations for the practical use of the results.
6. it is worth indicating which research gaps the authors fill. What additional, what has already been said in the literature has been added to it. What contributions the authors have made to research on the consequences of COVID.
7. it is also worth showing that the consequence of COVID is at least insecurity in several dimensions (social, health, but also what are the consequences for poverty).
8) It is also worth identifying the level of security of respondents.
9. importantly for me, I did not get an answer on how the questionnaire was obtained, how it was made realistic for the whole collective.
I consider the article itself worth pursuing further, as these are important scientific issues.
Author Response
We thank the reviewer for useful suggestions.
We addressed the comments as found below:
1) There is a lack of a thorough review of the literature; the problem has been addressed many times in the world, which the authors have not indicated.
We have cited relevant studies.
2) It would be useful to show how one moved from panic to denial of the problem. I myself had the opportunity to describe this phenomenon.
We thank the reviewer for this insightful comment. We have searched relevant literature to add this comment and reference [60], but it would have been helpful if the reviewer give a pointer to his/her work to get a comprehensive citation relevant to this work. If accepted, we are still willing to add this reviewer’s work.
3) It is worth noting that, using a fuzzy clustering method, it is possible to describe the threat states themselves, and to show whether and to what extent the problem should be addressed, but also which tools should be used.
Thank you for this excellent suggestion. First, we have limited knowledge of the application of this tool, and it seem late to apply for this particular work. However, we have added it in the limitation section and recommended its use in future works related to this. We added it as follows, ‘We are aware that using other excellent analytic tools like the fuzzy clustering method may better describe the present level of the threat and show whether and to what extent the problem should be addressed, while at the same time predict useful tools to address the problem [59]. We recommend the application of this method in future studies’.

Round 2
Reviewer 1 Report
The authors have improved their manuscript according to the suggestions. Kindly asking them to delete the "(%)" from line 173, column 3.
Author Response
Dear Reviewer,
Query: The authors have improved their manuscript according to the suggestions. Kindly ask them to delete the "(%)" from line 173, column 3.
Response: Thank you for noticing this oversight. We have now deleted it accordingly.
Thank you.
Reviewer 3 Report
Although the article has been partly completed, the literature review in the introduction is still missing. I myself have had the opportunity to write a number of articles on socio-economic security during pandemics. I then had the opportunity to look at publications in this field in the world literature (e.x. Aleksandra Łuczak or R. Śpiewak, Denysov, Kock, Kalinowski etc.). It is worth reviewing the research directions and pointing them out in the introduction.
There is also still a lack of recommendations and the summary is very laconic.
Author Response
We thank the reviewer for these useful comments and for pointing us to these new references.
Comment 1: Although the article has been partly completed, the literature review in the introduction is still missing. I myself have had the opportunity to write a number of articles on socio-economic security during pandemics. I then had the opportunity to look at publications in this field in the world literature (e.x. Aleksandra Łuczak or R. Śpiewak, Denysov, Kock, Kalinowski etc.). It is worth reviewing the research directions and pointing them out in the introduction.
Response: The new Lines 53-71 now addressed this query.
Comment 2: There is also still a lack of recommendations and the summary is very laconic.
Response: The new Lines 421-434 now addressed this query.
New references have been added and all references adjusted appropriately.